# Teachers' Social–Emotional Competence: History, Concept, Models, Instruments, and Recommendations for Educational Quality

**Gissela Lozano-Peña** [1] , **Fabiola Sáez-Delgado** [2,*] , **Yaranay López-Angulo** [3,4] and **Javier Mella-Norambuena** [5]

1    Doctoral Program in Education, Faculty of Education, Universidad Católica de la Santísima Concepción, Concepción 4330000, Chile; glozano@doctoradoedu.ucsc.cl
2    Centro de Investigación en Educación y Desarrollo (CIEDE), Faculty of Education, Universidad Católica de la Santísima Concepción, Concepción 4330000, Chile
3    Escuela de Psicología, Facultad de Ciencias Sociales y Comunicaciones, Universidad Santo Tomás, Concepción 4330000, Chile; yara13190@gmail.com
4    Departamento de Psicología, Facultad de Ciencias Sociales, Universidad de Concepción, Concepción 4330000, Chile
5    Departamento de Ciencias, Universidad Técnica Federico Santa María, Concepción 4330000, Chile; javier.mellan@usm.cl
*    Correspondence: fsaez@ucsc.cl

**Abstract:** Teachers' social–emotional competencies are essential to educational quality. This study aimed to describe the theoretical background and relevance of teachers' social–emotional competencies. We conducted a systematic review with a critical, theoretical review approach. The results showed that the concept has an increasingly complex history and followed a structured course from 1920 to present. Five main models have been identified: emotional regulation, prosocial classroom, Collaborative Association of Social Emotional Learning, Bar-On emotional intelligence, and emotional intelligence. There are measurement instruments consistent with four of the identified models; however, the model that does not have its own instrument uses different available scales. Specific recommendations are proposed to develop social and emotional competencies in educational public policies, which include school leadership, assessment, and teacher professional training. In conclusion, it is relevant to have clear guidelines that conceive and conceptualize social–emotional competence univocally. These guidelines would allow the design of instruments with a comprehensive and sufficient theoretical base that reflect the multidimensionality of the concept, provide a precise measure to assess the effectiveness of intervention programs, and enlist teachers who seek the development of the different skills that involve social–emotional competencies.

**Keywords:** socioemotional competence; models; emotional intelligence; social intelligence

## 1. Introduction

Social and emotional competencies (SEC) have been positioned as a central element in human development because of the high predictive capacity they have towards variables related to the educational context [1–3]. Thus, in recent years, international organizations, such as the European Union, the United Nations, and the OECD have recognized the relevance of SEC. This promotes their inclusion in international conventions and treaties signed with different countries [4]. In the same way, other organizations such as the World Bank, the World Health Organization, and UNICEF have joined efforts to establish a more explicit intention in the development of SECs [5].

The teaching profession is considered one of the most emotionally demanding professions, which can affect mental health and wellbeing [6,7]. It is also associated with episodes of stress and burnout [8]. A study conducted in Mexico [9] with 549 teachers from different educational contexts confirmed that SECs are predictors of burnout, and that teachers, in

general, have low emotional autonomy (M = 3.63; SD = 0.771); therefore, it is important to enhance teachers' personal skills to avoid personal and professional burnout. Another study conducted on 224 elementary school teachers in the United States [10] showed the effects of a program designed to improve teachers' stress by increasing awareness and resilience. The results showed significant decreases in psychological distress, pain-related reductions, physical discomfort, and a significant increase in emotion regulation and some dimensions of consciousness. The authors concluded that teachers who participate in SEC-related programs achieve a positive impact on their own wellbeing. Therefore, teachers' SEC has become relevant to their mental health, as confirmed by a systematic review on social and emotional learning interventions in teachers, which showed positive effects on wellbeing and psychological distress [11].

Teachers' SEC is considered to be a protective factor against stressful situations, in addition to promoting their wellbeing and their sense of self-efficacy in the classroom [12]. They are also relevant since teachers are the ones who execute social–emotional learning programs for students [6]. The literature shows vast evidence to consider SEC as a determining factor for improving educational quality [13–15], as they improve the teacher–student relationship and the classroom climate [3,16]. Teachers with high SEC establish positive relationships, provide support, and model these SEC for their students [17,18]. As a consequence of these positive effects, SEC indirectly improves students' academic performance [19,20].

Although the relevance of SEC of teachers and its contribution to educational quality is recognized, the understanding and delimitation of the SEC concept is an issue under discussion by researchers [21–23]. The multidimensional conformation of the construct that incorporates social, emotional, and other competency-based dimensions has implied an ambiguity regarding which of these three dimensions was the first to develop, to whom its development is attributed, and how they were incorporated into a single construct called SEC. In addition, there are difficulties at the inter- and intra-concept levels. At the interconcept level, SEC has been used interchangeably with other concepts, such as emotional intelligence. At the intraconcept level, inaccuracies are observed regarding the use of competencies, skills and abilities interchangeably, so that efforts are required to specify and clarify the inconsistencies in the literature [24].

Another issue discussed by researchers is related to the models on which the empirical studies related to teachers' SECs are based. These range from the use of theoretical models that consider emotional intelligence as a central concept, to others that incorporate more skills and specify SEC as a central concept [25,26]. This leads to a theoretical confusion in conducting research, due to the use of many variables and instruments when measuring SEC in teachers. With the background presented, having clear guidelines that conceive and conceptualize SEC univocally would allow the creation of instruments with a comprehensive and complex theoretical base that reflects the multidimensionality of SEC, avoiding the creation of subdefinitions in the field with instruments that partially measure SEC.

In the literature there are systematic reviews related to SEC, however, these are focused on students [27–29]; others are related to the field of health [30] and others have focused on constructs related to SEC, such as theoretical reviews on teacher emotional intelligence [31,32]. In definitive, no theoretical or systematic review focused on teachers' SEC has been identified. Considering that the knowledge gaps exposed refer to the theoretical and conceptual imprecision noted by the authors in the research on SEC, a comprehensive review of the literature is required to systematize the knowledge that has been developed on teachers' SEC. Therefore, the purpose of this study was to describe the theoretical background and support the relevance of SEC in educational quality. The study relied on the systematic review approach to answer the following research questions:

1.  What was the historical, conceptual, and theoretical path of the SEC construct?
2.  What are the models and measurement instruments for teachers' SEC?
3.  What recommendations are pertinent for the development of teachers' SEC as a way of contributing to educational quality?

This research will contribute to the knowledge by clarifying the theoretical–conceptual and empirical–methodological background regarding the teaching of SEC for future research, and provide new proposals to solve the challenges faced by teachers.

## 2. Methods

To answer the research questions of this study, a method that considered two stages was implemented. The first stage consisted of a systematic literature review to identify SEC models and instruments [33] and the second stage considered a theoretical review that included other investigations to account for the historical, conceptual, and theoretical path of the SEC construct [34], and to provide recommendations for the development of teachers' SEC as a way to improve educational quality.

### 2.1. Systematic Literature Review

This study was based on the guidelines, standards, and phases used by the available protocol for developing systematic literature reviews of PRISMA [33]. The studies that were identified have been analyzed based on the content-focused evidence information-management technique [35].

#### 2.1.1. Database and Concepts Used to Form the Search Algorithm

To identify the studies on teachers' SEC, an exploration of articles in the Web of Science (WOS), SCOPUS, and ERIC databases was conducted. These three databases have been selected since SCOPUS is the main database for peer-reviewed journals [36], the ERIC database is the main database for exclusive education studies [37], and the WOS database follows rigorous, internationally recognized, research quality standards. The search period was from 2010 to 2021. This time period was decided after an exploration of productivity on SES in the aforementioned databases, which showed an increase in productivity starting in 2010. The latest search and update date was 12 May 2021. The search considered articles where the concepts were incorporated in the title, abstract, and/or keywords of the studies. The concepts used were the different possibilities of the phrases "teacher" and "social–emotional competence." Thus, the search algorithm was formed as follows: ("teacher's social–emotional competence" OR "teachers' social–emotional competence" OR "emotional competencies" OR "emotional training" AND "teacher"). Once the search was conducted in each database, duplicates were eliminated; that is, studies that were in more than one database.

#### 2.1.2. Study Inclusion and Exclusion Criteria

In order to arrive at the most relevant sample of studies to be included in this review, five criteria were defined: (1) in English, Spanish or Portuguese, (2) empirical research, (3) in a school context, (4) focused on teachers (5) complete and accessible manuscript. This research, based on the systematic review method, only included empirical studies to answer question 2, which consists of identifying the models and measurement instruments for teachers' SEC applied over the last 10 years in the research.

#### 2.1.3. Search Process Results

The search process helped identify a total of 95 types of research in the consulted databases (Figure 1). Then, 35 study records were deleted because they were in duplicate or automatically selected by the automation software. As part of the first screening process, of the 60 studies that passed to this phase, two of the authors reviewed the titles and abstracts to ensure that the studies had teachers' SEC as a central theme and met the established inclusion criteria. Disagreements that appeared between the two authors regarding the selection or not of a study, after analyzing its title and abstract, were solved with a discussion involving all the authors. This first screening process resulted in the selection of 44 articles. The second screening process, which consisted of the complete reading of the texts of each study, applied the 4 inclusion and exclusion criteria that were

defined a priori. A total of 29 studies were excluded, and consequently, 15 studies were selected (see Figure 1).

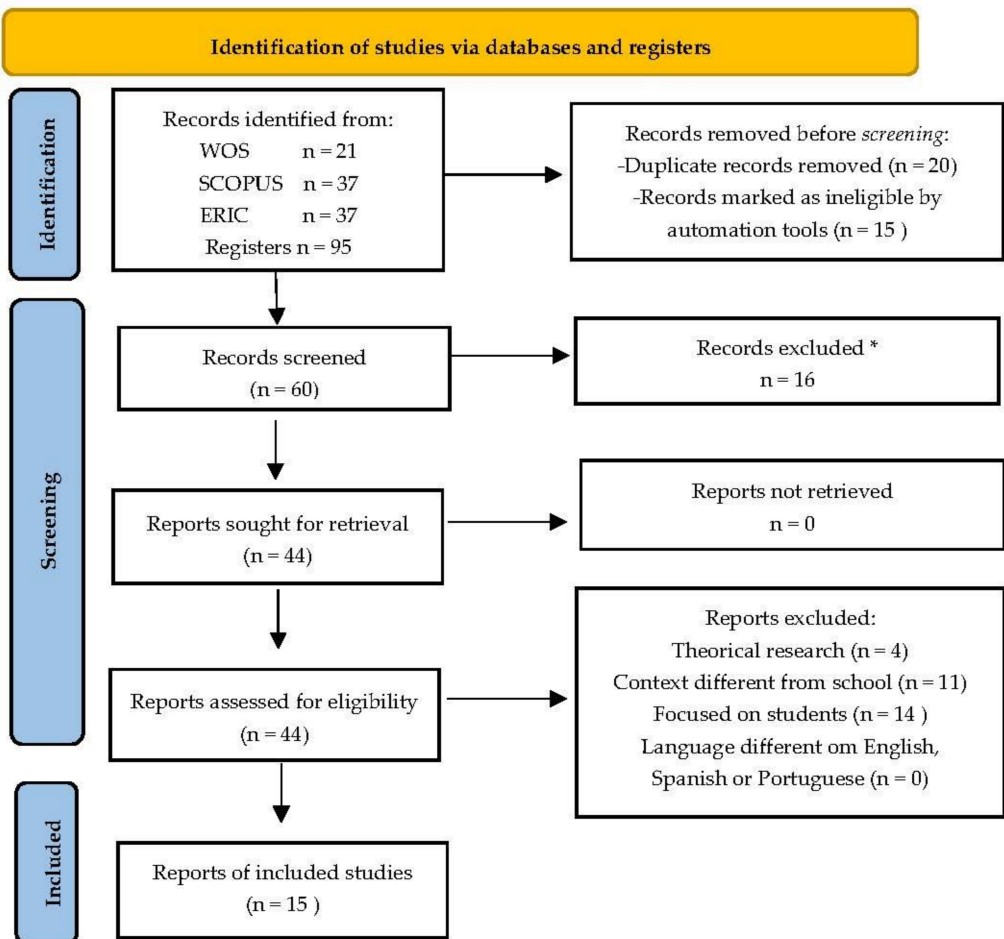

**Figure 1.** PRISMA flow diagram of the search and selection process for articles about teacher's SEC [33], * records excluded, without teachers' SECs concept in the title, keywords or abstract.

2.1.4. Study Content Analysis Process

The content analysis process of the studies followed two stages. Based on the 15 empirical studies included in this review on teachers' SEC, the first stage of content analysis consisted of reading each text of the studies carefully and meticulously to identify the theoretical models that the authors used to support their research. A matrix was constructed for extraction and systematization of the information (See Appendix A, Table A1). To validate the information extracted from the 15 studies, two of the four researchers extracted the content independently. Only when there were discrepancies, a third researcher extracted the content independently. The final information was agreed upon with all the authors. The second stage of content analysis was to go to the original source of each of the teachers' SEC models identified in empirical studies in the last decade (2001–2021), to describe them, and answer the research questions of the present review.

*2.2. Theoretical Review*

The second part of the method was based on a theoretical review with emphasis on the narration of the conceptual and chronological aspects [34], which sought to complete the findings of the systematic literature review to answer research questions 1 and 3 of this study. It consisted of a selection of theoretical articles and key-book chapters that allowed us to establish and describe a historical view of the origin of the SEC construct. To present the conceptual evolution of the construct, empirical studies that have presented a

definition of the construct were taken as a reference and allowed us to present its evolution from 1997 to 2020. Likewise, a search process for the SEC concept has been conducted in the SCOPUS, WOS, and ERIC databases without a year limit to analyze the scientific production available for each year until 2021.

## 3. Results

Next, the results of this study are presented to answer the established research questions regarding the historical and conceptual path of the SEC construct, its theoretical models and instruments, and some recommendations for developing teachers' SEC as a way to contribute to the quality of education.

### 3.1. Historical and Conceptual Path of the SEC Construct

3.1.1. Historical SEC Path

The result of the SEC concept path is presented from two perspectives. The first represents the path from the review of the specific literature of the field and the second from the frequency of productivity of scientific articles evidenced in the WOS, SCOPUS, and ERIC databases.

Regarding the first perspective to account for the historical path, although it is not possible to determine a single temporality of the development of the components that make up the SEC, an emphasis on different periods can be identified over time and four periods can be identified: (1) The beginning of the concept centered on the social component, (2) the beginning of the concept centered on the emotional component, (3) the beginning of the concept with integration of the social and emotional components, and (4) the beginning of the concept that understands the social and emotional component as competence.

In 1920, Thorndike was credited with the concept of social intelligence, and the relevant instrument was developed by Moss and Hunt; however, the measurement results of this instrument were unsatisfactory [38]. These results, together with the rise of behaviorism, diminished research related to social intelligence for a time, as can be seen in Figure 2 [39]. However, the intelligence model of the intellect structure proposed by Guilford and Bandura's theory of social learning appears to establish that there is a combination of social and psychological factors that influence behavior [40].

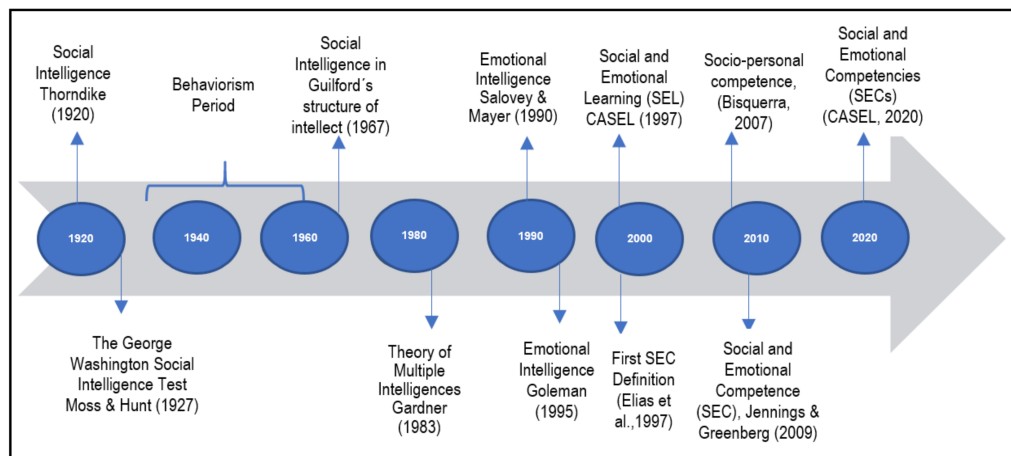

**Figure 2.** A timeline in the understanding of the evolution of the socioemotional competence concept. Source: [17,21,38,40–42].

Later in 1983, Gardner proposed the theory of multiple intelligences by establishing intrapersonal and interpersonal intelligence [39], which were the basis for defining emotional intelligence proposed by Mayer and Salovey, and were understood as the ability to observe one's feelings and emotions as well as those of others, to distinguish between these

emotions and to use this information to direct action and thought [43], a concept made popular by Goleman [39].

The confluence of the social and emotional components is appreciated for the first time in the social–emotional learning construct developed by nine Collaborative Association of Social Emotional Learning (CASEL) collaborators [41], and is defined as the process through which people acquire and effectively apply the knowledge, attitudes, and skills that are required to understand and manage emotions, establish and achieve positive goals, show empathy for others, establish and maintain positive relationships, and make responsible decisions [7].

Finally, in the literature review, the first definition of SEC which includes competence, was the one established by Elias [41], defined as the ability to understand, manage, and express social and emotional aspects of people for success in the development of tasks, learning, relationships with others, problem solving, and adjustment to the demands of the context.

Regarding the second perspective which accounts for the historical path, Figure 3 presents all the publications per year registered in SCOPUS and WOS with the search for the SEC concept. The purpose of this exploration was to identify the first time the SEC construct appears in the databases; therefore, the search was not limited to a specific time range. In addition, this search allows us to identify the frequency of studies on this topic by year.

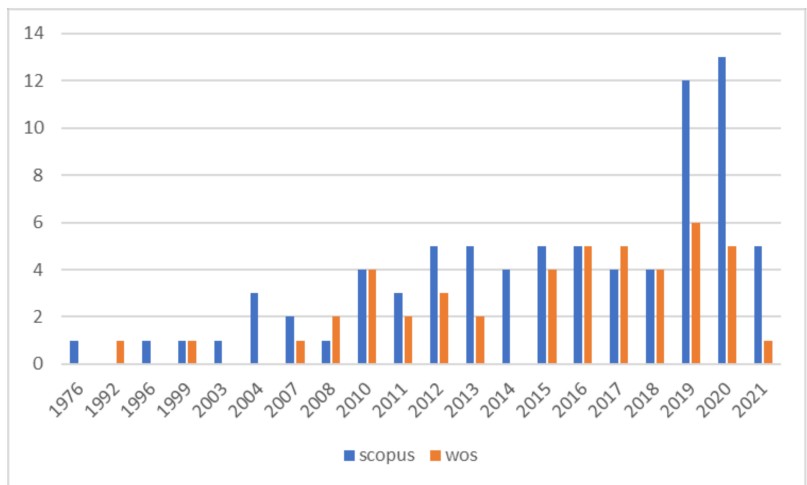

**Figure 3.** Number of publications indexed in SCOPUS and WOS databases of SEC concept.

In the ERIC database, productivity is presented by time range; for this reason, it was not included in the graph. In addition, it shows productivity from 2002 onwards since from this date it is part of the new Institute of Sciences of Education [44]. Given these characteristics of the ERIC database, 47 studies were observed ranging from 2002 to 2011, 79 studies between 2012 and 2016, 72 studies ranging from 2017 to 2019, and 40 ranging from 2020 to 2021. The first study published on this basis in 2002 was entitled Showing and telling about emotions: Interrelations between facets of emotional competence and associations with classroom adjustment in Head Start preschoolers, developed by Miller [45].

Regarding the SCOPUS database, the first registered publication was entitled Students' Perceptions of Female Professors, research carried out by Mackie [46], although the SEC concept appears in the research summary, this is not the central theme of the study. After a period of 20 years, a second publication appears, Behavioral Assessment of Coping Strategies in Young Children At-risk, Developmentally Delayed and Typically Developing, developed by Stoiber and Anderson [47]. Like the first publication, it does not develop the SEC concept as the central theme of the study. Between 1996 and 2008, a reduced number of publications, with nine studies, was observed. As of 2010 and until 2018, a constant number of publications is maintained, ranging from three to five records per year. However, in the

last two years 2019-2020, there has been a significant increase in publications related to SEC, with 12 and 13 studies, respectively. Regarding the WOS database, the first publication is developed by Anderson in 1992 [48] with the title Effects of day care on cognitive and socioemotional competence of 13-year-old Swedish schoolchildren, incorporating SEC as one of the central themes of the article. Between 1999 and 2007, only two publications were registered. Between 2008 and 2013, 13 publications were observed. Finally, a greater frequency is observed in scientific productivity on the subject of SEC in recent years from 2015, reaching 30 publications. It can be seen that ultimately, the frequency of productivity per year increases significantly as of 2010.

3.1.2. Definition Path of the SEC Concept

In the available literature, there are various SEC conceptualizations that have been conceived in their first approaches to the concept as the skills, motivations, knowledge, or abilities that a person has to face and master in social and emotional situations with a certain level of efficiency and quality [21,41]; until arriving at more recent conceptualizations such as the effective management of intrapersonal and interpersonal social and emotional experiences, and promoting prosperity and wellbeing, as well as that of others [49]. Considering the diversity of conceptualizations available in the literature, some definitions have been evaluated that allow us to establish an evolution of the concept over the years, as shown in Table 1.

**Table 1.** Evolution of SEC definition.

| Year | Social–Emotional Competence Definition |
|---|---|
| 1997 | Social–emotional competence refers to a person's knowledge, skills, and motivation required to master social and emotional situations. |
| 2002 | A multivariate concept that includes a person's ability to identify their emotions, to be able to manage their emotions appropriately, to have positive interactions, and to have positive interactions with others. |
| 2003 | A set of social and emotional skills to achieve a goal both in the personal and professional spheres. |
| 2007 | The ability to appropriately mobilize a set of knowledge, skills, abilities and attitudes to perform different activities with a certain level of quality and efficiency. |
| 2009 | A comprehensive set of interrelated skills and processes, including emotional processes (e.g., understanding and regulating emotions, taking others' perspectives, recognizing their own emotional strengths and weaknesses), social and interpersonal skills (e.g., understanding social cues and interacting positively with others), and cognitive processes (e.g., stress management, impulse control). |
| 2011 | A multidimensional concept, cognitive, attitudinal and behavioral, and it involves uncertainty. |
| 2012 | Knowledge, skills and social and emotional attitudes, put into practice in real life. |
| 2013 | Teacher SEC is understood as a comprehensive set of interrelated skills and processes, including emotional processes (e.g., understanding and regulating emotions, taking others' perspectives, recognizing their own emotional strengths and weaknesses), social and interpersonal skills (e.g., understanding social cues and interacting positively with others), and cognitive processes (e.g., stress management, impulse control.) |
| 2017 | Skills, knowledge, attitudes, and social and emotional dispositions that enable a person to set goals, manage behavior, build relationships, and process information in diverse contexts that intentionally develop these competencies. |
| 2019 | Teacher SEC is defined in terms of the five competencies: self-awareness, self-management, social awareness, relationship skills and responsible decision making. |
| 2020 | Effective management of intrapersonal and interpersonal social and emotional experiences in ways that foster one's own and others' thriving. SEC is operationalized by individuals' social–emotional basic psychological need satisfaction, motivations, and behaviors. |

Source: [10,49–57].

Based on the above-described conceptualizations, it is possible to identify various constructs used to refer to SEC, capabilities, skills, knowledge, attitudes, experiences, abilities, etc. The skills construct has been quite controversial, as it is a polysemic concept since its origin and development can be attributed to multiple disciplines and contexts [58],

so there is no single concept. However, there is a consensus in considering ability as having the potential to learn (cognitive, affective, psychomotor); skill refers to knowing how to perform an action, and competence refers to taking actions (performing) with excellence. Capacity, as shown in Figure 4, refers to the basic resources that a person possesses, which have a biological basis [59]. Ability refers to having the potential to learn cognitive, affective, psychomotor skills, etc. Abilities are skills and behavior, that is, they are developed capacities; the skills go one step further, that is, they are the skills used flexibly, correctly, and appropriately in various contexts, that is, to perform actions or perform with excellence [60].

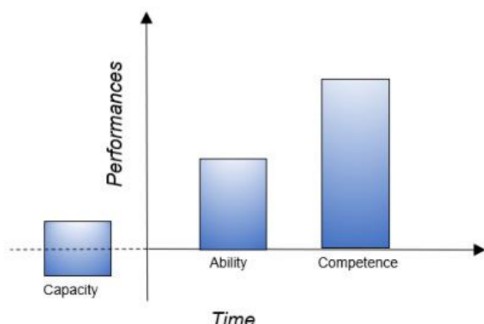

**Figure 4.** Disambiguation of competence concept [60].

A relevant conceptualization suggests that the concept of competence is the combination of the cognitive, motivational, moral, and social skills that a person or a social group possess or can learn, that underlie successful mastery through the understanding and appropriate actions of a variety of demands, tasks, problems, and goals [61]. They are the basic interpersonal, strategic, and execution skills [62]. According to the conceptual analysis and distinction of the concepts of capacity, ability, and competence and assessing the definition proposed by Collie [49], SEC could be defined as the effective deployment of abilities that allows subjects to cope with social and emotional intrapersonal and interpersonal experiences assertively.

### 3.2. Theoretical SEC Models and Instruments

#### 3.2.1. Theoretical SEC Models

Five theoretical models that met the inclusion criteria have been selected to be analyzed in the present study (See Appendix A, Table A2): Gross' model of the emotional regulation process in 1998 [63]; Mayer and Salovey's emotional intelligence model in 1997 [25]; Bar-On's emotional intelligence model in 1997 [64]; Jennings and Greenberg's 2009 prosocial classroom model [17]; and the CASEL social–emotional learning model in 2013 [26]. The models are characterized below.

Emotion Regulation Process Model

Gross' model is based on emotional regulation, understood as the processes by which people influence the emotions they have, and how they experience and express them. These emotion regulation processes can be automatic or controlled, conscious or unconscious, and their effects can be shown at one or more points in the emotion generation process [65].

This emotion regulation process model facilitates and allows the analysis of types of emotion regulation by establishing five sets of emotional regulatory processes as shown in Figure 5: situation selection, situation modification, attention deployment, cognitive change, and response modulation. This is an elaboration of the two-way distinction be-tween antecedent-centered emotion regulation, a pre-emotion process, and response-centered emotion regulation, a process that occurs after the emotion is generated [63,66].

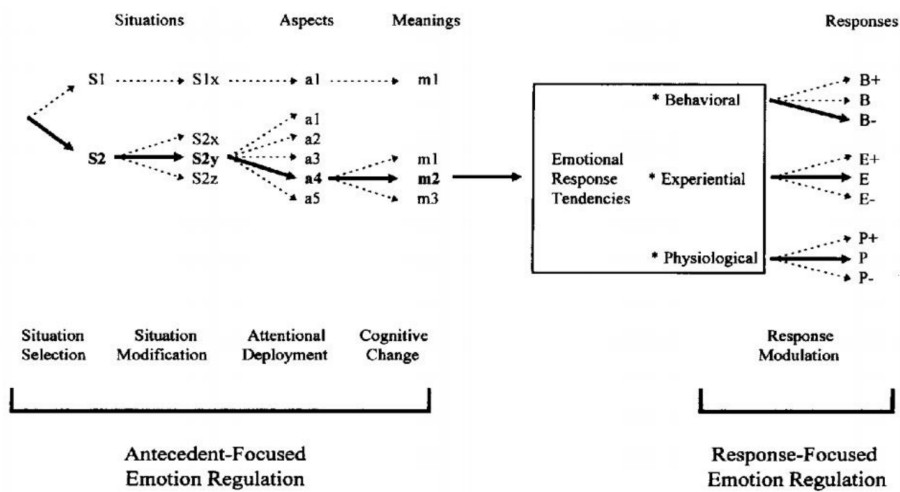

**Figure 5.** A process model of emotion regulation [63].

Mayer and Salovey's Emotional Intelligence Model

The Mayer and Salovey ability model, graphically represented in Figure 6, considers emotional intelligence as a concept and conceptualizes it through four basic skills, which are: perceiving and expressing emotions, accessing and/or generating feelings that facilitate thinking; understanding emotions and emotional awareness and regulating emotions promoting emotional and intellectual growth [25]. In this way, these four basic skills are what make this model a skill model, as proposed by Trujillo and Rivas [39] who also classify it as a skill model.

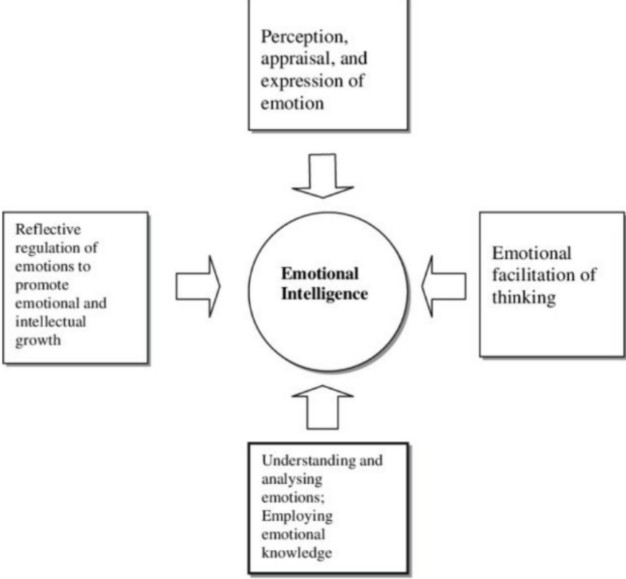

**Figure 6.** Schematic presentation of Mayer and Salovey's 1997 model of emotional intelligence [67].

Bar-On's Emotional Intelligence Model

According to this model, emotional intelligence is a representative set of skills and emotional and social facilitators that interrelate and determine the effectiveness with which a subject understands and expresses himself, understands and relates to others, and efficiently deals with daily demands [64].

Among the dimensions covered by the model as shown in Figure 7 is development: intrapersonal, interpersonal, stress management, adaptability, and general mood. The social components of this model are the interpersonal and adaptability dimensions, and the intrapersonal emotional components are stress management and general mood.

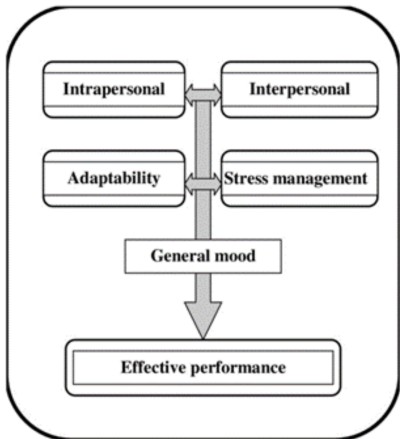

**Figure 7.** Bar-On model of emotional intelligence [64].

Prosocial Classroom Model

The Jennings and Greenberg prosocial classroom model has social and emotional skills as its central concept. This skill uses the definition developed by CASEL [26] which involves five main emotional, cognitive, and behavioral skills: self-awareness, social awareness, responsible decision making, self-management, and relationship management.

As can be seen in Figure 8, the prosocial classroom model is structured into five dimensions: social and emotional skill and teacher wellbeing, teacher–student relationships, classroom management, implementation of the social and emotional learning program, and, finally, classroom climate. This model emphasizes the importance of these five dimensions in creating a climate favorable to learning in the classroom and in promoting positive results in student development [17].

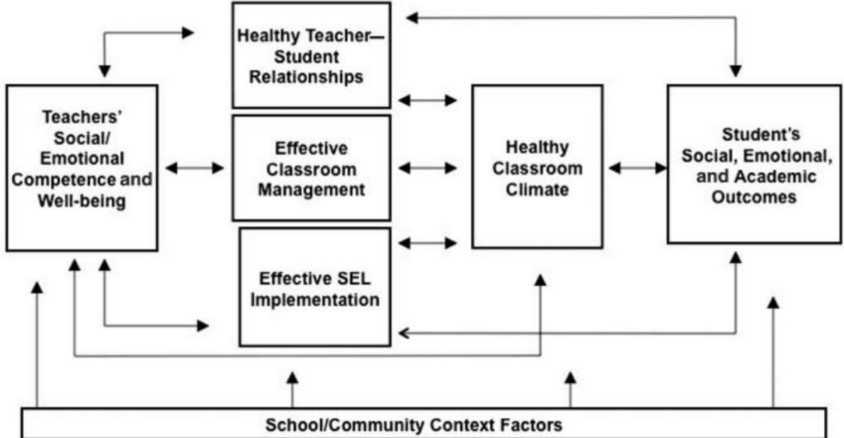

**Figure 8.** The prosocial classroom: A model of teachers' social and emotional competence and classroom and student outcomes [17].

The model proposed by Jennings and Greenberg raises social and emotional components. A predominant social component is observed since we find it in its five dimensions. To a lesser extent, it presents an emotional component in two of its dimensions, the teacher's social and emotional competence and wellbeing, and the implementation of the social and emotional learning program respond to the emotional component.

CASEL Model

For its part, the CASEL model establishes as a basis the concept of social and emotional learning, defining it as learning that involves processes through which children and adults acquire and develop knowledge, skills, and attitudes that are needed to understand and

manage emotions, as well as to achieve positive results, meet goals, demonstrate empathy, maintain positive relationships, and make responsible decisions [26].

Figure 9 represents the first image of the CASEL model, which, despite having been updated, maintains the same skills. This figure raises five interrelated sets of cognitive, affective, and behavioral skills: self-awareness, self-management, responsible decision making, relationship skills, and social awareness. The framework emphasizes the importance of establishing equitable learning environments and coordinating practices in classrooms, schools, families, and school communities to enhance students' social, emotional, and academic learning [67].

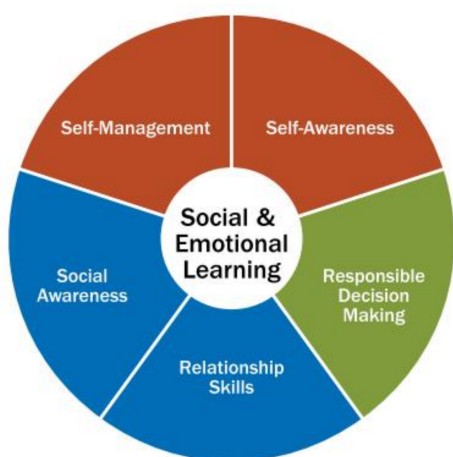

**Figure 9.** The five social and emotional learning core competencies [26].

The CASEL model presents social components such as self-management, relationship skills, and social awareness. Likewise, it establishes emotional components such as self-awareness and responsible decision making.

Ultimately, Table 2 synthesizes and represents an extension of the characteristics of the theoretical models on which the research is based on teachers' SEC. The models have been classified in relation to the central concepts and skills that comprise them. Three types of models have been established: emotional intelligence models, among which are the Mayer and Salovey's and Bar-On's models; emotional regulation models, including the Gross' model; and the models of social and emotional development among were identified Jennings and Greenberg's prosocial classroom and CASEL's social–emotional learning models. Although each model presents a different theoretical proposal, all these models consider between four to five integrative skills related to the social and/or emotional field. Regarding the number of citations, the Mayer and Salovey model of emotional intelligence is the most cited in the literature.

### 3.2.2. SEC Measurement Instruments

Of the five models analyzed in this review, four of them have an instrument consistent with their integrative skills: (1) Gross' model with its emotion regulation questionnaire; (2) the Bar-On's model with its emotional quotient inventory; (3) the Mayer and Salovey's model with its Trait Meta-Mood Scale self-report measure (MSCEIT) performance measure; (4) and the CASEL model with Yoder's Social and Emotional Skills Questionnaire [68,69]. On the other hand, Jennings and Greenberg's model does not have a coherent instrument for assessing its integrated skills. However, the investigations that are based on this model, to empirically measure their integrative skills, use other instruments, thus using more than one, as shown in Table 3.

According to the population of users of the models for conducting research, most of them have been used in adults, children, and adolescents. Meanwhile, Jennings and Greenberg's model has been used mainly in adults.

**Table 2.** Synthesis table of theoretical models on SEC.

| Emotional Intelligence Models | | Emotional Regulation Model | Social–Emotional Development Models | |
|---|---|---|---|---|
| Mayer and Salovey (1997) Emotional intelligence model | Bar-On (1997) Bar-On emotional intelligence model | Gross (1998) Emotional regulation process model | Jennings and Greenbergs (2009) Prosocial classroom model | CASEL (2013) Social and emotional learning model |
| **Definition** Emotional intelligence: is a set of abilities that account for how people's emotional perception and understanding vary in their accuracy. More formally, we define emotional intelligence as the ability to perceive and express emotion, assimilate emotion in thought, understand and reason with emotion, and regulate emotion in the self and others. | **Definition** Emotional intelligence: is an array of noncognitive capabilities, competencies, and skills that influence one's ability to succeed in copying with environmental demands and pressures. | **Definition** Emotional regulation: is defined and distinguished from coping, mood regulation, defense, and affect regulation. Emotion is characterized in terms of response tendencies. | **Definition:** Social and emotional competence: use the broadly accepted definition of social and emotional competence developed by CASEL. This definition involves five major emotional, cognitive, and behavioral competencies: self-awareness, social awareness, responsible decision making, self-management, and relationship management. | **Definition:** Social and emotional learning: involves the processes through which children and adults acquire and effectively apply the knowledge, attitudes, and skills necessary to understand and manage emotions, set and achieve positive goals, feel and show empathy for others, establish and maintain positive relationships, and make responsible decisions. |
| **Major areas of skills** Perception and expression of emotion Assimilating emotion in thought Understanding and analyzing emotion Reflective regulation of emotion | **Major areas of skills** Intrapersonal skills Interpersonal skills Adaptability scales Stress-Management scalesGeneral Mood | **Major areas of skills** Situation selection Situation modification Attentional deployment Cognitive change Response modulation | **Major areas of skills** Teacher's social–emotional competence and wellbeing Teacher–student relationships Effective classroom management Social–emotional learning program implementation Classroom climate | **Major areas of skills** Self-awareness Self-management Responsible decision making Relationship skills Social awareness |
| **n° citation** 12,606 | **n° citation** 2015 | **n° citation** 8926 | **n° citation** 2935 | **n° citation** 55 |

Source: [17,25,26,63,64].

**Table 3.** Models, measuring instruments and population.

| Model | Measuring Instrument | Population |
|---|---|---|
| Emotional regulation process model [63] | Emotion Regulation Questionnaire (ERQ) Dimensions:<br>(a) Cognitive reappraisal<br>(b) Expressive suppression | Adults Children and teenagers |
| Prosocial classroom model [17] | Different instruments for measuring SEC, example:<br>1. Interpersonal Reactivity Index<br>2. 2TSEC perception scale | Adults |
| Social and emotional learning model [26] | Socioemotional competence questionnaire Dimensions:<br>(a) Self-awareness<br>(b) Self-management<br>(c) Responsible decision making<br>(d) Relationship skills<br>(e) Social awareness | Adults Children and teenagers |

**Table 3.** *Cont.*

| Model | Measuring Instrument | Population |
|---|---|---|
| Bar-On Emotional intelligence model [64] | Bar-On EQ-I Dimensions<br><br>(a) Intrapersonal<br>(b) Interpersonal<br>(c) stress management<br>(d) Adaptability<br>(e) General mood | Adults<br>Children and teenagers |
| Emotional intelligence model [25] | (a) Self-report measure:<br><br>Trait Meta-Mood Scale Dimensions:<br><br>(1) Attention<br>(2) Clarity<br>(3) Repair<br><br>(b) Performance measurement:<br><br>MSCEIT Dimensions<br><br>(1) Perceiving and expressing emotions<br>(2) Using emotions<br>(3) Understanding emotions<br>(4) Regulating emotions | Adults<br>Children and teenagers |

*3.3. Recommendations for Developing Teachers' SEC as a Way to Contribute to Educational Quality*

Understanding SEC from theoretical, conceptual, and empirical aspects contributes to making better decisions in the research and educational field. Therefore, below, some recommendations are proposed for the development of SEC as a contribution to educational quality in (a) the evaluation of SEC at the school and public policy level, (b) teacher training in SEC; and (c) the leadership of educational institutions.

3.3.1. Recommendations for Assessing SEC in Teachers and Their Students

First, it should be noted that SEC can be taught, learned, assessed, and trained [56,70,71]. In this sense, less progress has been made in developing methods to assess social and emotional skills in school [71]. Therefore, it is necessary to establish the SEC assessment in both students and teachers. In relation to the teacher, the use of instruments that allow us to assess their SEC in the classroom should be improved, as well as this, teachers should be trained in the construction of social–emotional learning assessments in order to use them constructively in the assessment of their students [72]. Regarding the student, progress must be made with evaluations specifically designed to measure their knowledge, ability, and social and emotional disposition during interpersonal interactions and their participation in school and community life [73].

Assessing the SEC in teachers and students can also be developed at a standardized level, which would imply consideration of its measurement at the public policy level as an index of educational quality, for example, the incorporation of tests (in students) or certifications (in teachers) at the country level.

3.3.2. Recommendations for Teacher Training in SEC

SECs are malleable compared to IQ; therefore, they can be trained through interventions, including in adulthood [6]. In this sense, if one wants to advance towards a trans-versal development of SEC in education, the incorporation of an explicit approach in teacher training is unavoidable [18]. In this sense, there seems to be a deep disconnect be-tween the skills that teachers require to develop social and emotional learning in schools and what the teacher training universities offer them [74]. This challenge should be raised on three levels: First, in teacher training programs, subjects that cover their entire period of study training, and that allow them to develop and train their SEC, should be incorporated.

Second, there should be SEC development, follow-up, and training of in-service teachers, who probably lack these SECs, and who have probably had to use personal resources to develop social and emotional learning programs with their students. A third level is related to the need for joint work between teacher training entities and school communities to develop and assess SEC in teachers in training and in service.

### 3.3.3. Recommendations to Strengthen the Leadership of Educational Institutions

Educational leaders have a direct impact on educational quality. The existence of a strong relationship between directive leadership in a school and the achievement of student learning results has been evidenced, in fact, the leader of a school is considered to be the second factor, after the teacher, with the greatest influence on the student academic achievement [75,76]. Given the relevance and impact that a school leader generates, SEC learning or training should be incorporated into the prior and ongoing professional learning of educational leaders [71].

Recently, leadership for social justice has gained notoriety, which aims to address the complexity of highly vulnerable schools [77]. In these contexts, it is essential to exercise leadership that recognizes that SEC, in addition to being a predictor of behaviors or positive results at the school level, acts preventively, as protection against risk factors that can harm students [78], such as problems of violence, delinquency, substance use, and dropout rate [79–81]. Leadership that recognizes that the SEC and wellbeing of the teacher influences the learning context, the implementation of social and emotional learning programs, and the relationships that exist between teachers and students.

## 4. Discussion

This review aimed at describing the theoretical background and supporting the relevance of SEC in educational quality by; (1) elucidating historical, conceptual, and theoretical aspects of the SEC construct; (2) identifying models and instruments for measuring SEC; and (3) proposing guidelines for developing SEC as a way to contribute to educational quality. Next, the results are discussed in relation to objectives, educational implications, and limitations, and projections for future research are proposed.

Concerning the historical path of the SEC concept, this study showed an increasingly complex and structured course. Since it is inherent to human intelligence, it becomes more complex with contextual and social development, and poses challenges to decision making and the search for problem solving. Thus, the discussion and reflection of the research regarding intelligence other than the cognitive aspect of the IQ of the human being began with the social intelligence of Thorndike (1920), followed by several other relevant theoretical contributions that contributed to the configuration of the theory of emotional intelligence. This finding is consistent with that indicated by Joseph and Newman [82] who point out that emotional intelligence is the embodiment of the concept of social intelligence.

The study also recognizes that the emotional intelligence theory has been crucial in understanding the SEC construct. This result is consistent with that established by Mikulic and other researchers [83], who warn that advances in emotional intelligence have con-tributed to the delimitation of the SEC construct.

From the results of this study, with regard to the conceptualizations, it can be pointed out that there is a great diversity of SEC definitions. However, it is also possible to observe that these definitions share common elements, favoring the idea that SEC can be observed from a three-dimensional perspective, and configured by a social, emotional, and competence component. Based on this background and framing this concept in the field of school-level teachers, teachers' SECs have been defined as the effective deployment of skills that allows teachers to function in social and emotional, intrapersonal and interpersonal experiences assertively in the educational context.

The results of this study also evidenced the existence of five theoretical models that empirical research uses as a theoretical basis in studies that address SEC with a sample of school-level teachers. Specifically, two of these, corresponding to the Bar-On's and Mayer

and Salovey's models, are directly supported by the theory of emotional intelligence, since this is their central concept [25], therefore, they were classified as models of emotional intelligence. In relation to Jennings and Greenberg's prosocial classroom, and that of CASEL social–emotional learning, are more inclusive, since they consider learning and/or social and emotional skills to be central concepts and include social and emotional skills, which are called integrative skills, therefore, these two were classified as social and emotional development models. Finally, Gross' emotional regulation Model was focused on the emotional dimension of SEC.

The emotional intelligence models developed by Bar-On and Mayer and Salovey seem to have been designed from their central concept and skills, as models applicable to any type of context, work, educational or organizational setting, etc., while the models of Jennings and Greenberg and CASEL are models specifically applicable to the educational and/or school environment. In relation to the skills or integral variables of the models, these are quite broad and vary according to authors and models, and this finding is consistent with other studies [23,84]. Although some of these models have been built to contribute specifically to the educational field, they focus on how to generate SEC in students but neglect the development of the application of these models in teachers.

Regarding the measurement instruments, although the four models present coherent and consistent instruments between the skills or variables, it can be observed that the only instrument that explicitly measures SEC is the Yoder social and emotional competence questionnaire, designed and used in the research for the CASEL framework. This instrument allows for the comprehensive measurement of both social and emotional skills, in addition to the purpose that it establishes in its description. Another relevant point is that most of the instruments are able to self-report, which could have the risk of spurious correlations as a consequence of the common bias of the method, with the participants reporting on their social and emotional skills or other results [85]. An exception is an instrument established for Mayer and Salovey's model, which in addition to its self-report measure, the Trait Meta-Mood Scale, presents a performance measure with the MSCEIT. Researchers have warned of the need to continue improving the current ways of measuring teachers' SEC, moving towards the development of instruments to empirically investigate what kind of knowledge and skills teachers should acquire, and the construction of instruments that overcome the barrier of self-reports [86,87].

The limitations of this study consider aspects related to the method. The literature re-view for the first stage of this study considered exploration in only three databases. Another limitation is that only studies published in English, Spanish, and Portuguese have been selected, eliminating those in other languages. Another limitation of the study is related to the sample size of the systematic literature review, with 15 articles resulting from the search in the last 10 years, even though before this period there were some studies that were left out of the analysis.

One of the main strengths of this theoretical study is that it has used a method that complements two techniques, one of systematic literature review and the other of critical review. This has made it possible to account for a historical journey of the SEC concept, identify and analyze the models most used in empirical investigations of teachers, expose the instruments that measure SEC, and specify recommendations for developing SEC as a contribution to educational quality.

Future studies can contribute to the proposal of a solid theoretical model, based on the analysis for developing teachers' SEC. It is also important to consider the development of empirical studies that describe teachers' profiles regarding the SEC levels. This would help identify which variables are related to a high level of SEC at the level of teachers and students. Multilevel interventions should also be developed that evaluate the effectiveness of the training for the improvement of SEC, not only in teachers who are exposed to these trainings, but also with measurement in the student body, to show evidence of how a teacher with high levels of SEC can influence the promotion of these skills in their students and in variables such as the classroom climate, academic performance, and dropout rate.

In our current society, social and educational change processes are constantly transforming, conditioning and stressing the teaching profession. Undoubtedly, the COVID-19 pandemic has left an enormous challenge in the educational field, increasing the need to develop SEC in teachers to improve their emotions, relationship with others and to support students emotionally. In this context, future research should consider how to train, coach or improve SEC in both, in-service and preservice teachers.

Including SEC in the educational field has mainly focused on the implementation of social and emotional learning programs aimed at students [5,88]. In this way, it is necessary to advance and incorporate social and emotional learning in the different educational public policies that include school leadership, assessment, and professional teacher training. [5]. This will allow contribution to a new area that supports the improvement of educational quality in schools.

In conclusion, this study contributes to academia and education. In the academic aspect, this study presents to researchers the historical path of the SEC concept and its configuration from different theoretical approaches showing its evolution. It opens the field for the proposal of new theoretical models focused on how to develop the three-dimensionality of the SEC concept, framing it specifically to the research field of teachers. Regarding the contribution to education, this study goes beyond socioemotional learning centered on students, and makes visible the importance of SEC in teacher training. Furthermore, this study makes it possible to rethink education as something that considers teachers as professionals who need to be trained academically and socioemotionally in order to achieve better education and a better society.

**Author Contributions:** Conceptualization, G.L.-P. and F.S.-D.; methodology, F.S.-D. and J.M.-N.; formal analysis, G.L.-P. and F.S.-D.; writing—original draft preparation, G.L.-P., F.S.-D., Y.L.-A. and J.M.-N.; writing—review and editing, J.M.-N.; supervision, F.S.-D.; project administration and funding acquisition, F.S.-D. All authors have read and agreed to the published version of the manuscript.

**Funding:** This work received funding from the National Research and Development Agency of the Chilean Government [ANID, Proyecto FONDECYT Iniciación 11201054] and Scholarship Program/Doctorado Nacional Chile/2020-21202422.

**Institutional Review Board Statement:** Not applicable.

**Informed Consent Statement:** Not applicable.

**Data Availability Statement:** Further inquiries can be directed to the corresponding author/s.

**Conflicts of Interest:** The authors declare no conflict of interest.

## Appendix A

**Table A1.** Extraction Information Matrix (SLR).

| ID | Citation | Participants Characteristics | Socioemotional Competence Definition/or Similar Concept | Theoretical Model | Approach, Design and Sample | Instruments | Instruments Dimensions |
|---|---|---|---|---|---|---|---|
| 1 | Aldrup et al. [50] | (a) Germany (b) Secondary (c) Pre/in service teachers | Social–emotional competence refers to a person's knowledge, skills, and motivation required to master social and emotional situations. | Emotional regulation process model (Gross, 1998) | (a) Quantitative (b) Correlational (c) 346 | 1. Test of regulation and understanding of social situations in teaching (TRUST) (Authors elaboration) | (1) Emotional regulation (2) Relationship management |
| 2 | Brown et al. [51] | (a) USA (b) Primary-secondary (c) In-service teachers | Teachers' SECs include a set of five interrelated skills: self-awareness, social awareness, self-management, relationship skills, and responsible decision making. | Social and emotional learning model (2013) | (a) Mixed (b) Not available-Correlational (c) 76 | 1. Semi-structured interviews 2. Socioemotional competence questionnaire. | (1) Self-awareness (2) Social awareness (3) Relationship management (4) Responsible decision making (5) Self-management |
| 3 | Buzgar and Giurgiuman [89] | (a) Romania (b) Primary-secondary (c) In-service teachers | Social–emotional learning refers to the process through which children and adults acquire and efficiently apply knowledge, attitudes and abilities in order to understand and control emotions, establish and achieve personal goals, feel and express empathy towards others, maintain positive relations with people, and make responsible decisions. | Not available | (a) Mixed (b) Correlational-grounded theory (c) 120 | 1. Questionnaire designed by authors | (1) Students' age (2) Teacher's expertise (years) (3) Teacher's county (4) Teacher's SEL training (5) Socioemotional learning program |
| 4 | Cheng [90] | (a) China (b) Primary-secondary (c) In-service teachers | Emotional competency is the social and emotional ability to cope with the demands of daily life. It determines how effectively individuals understand and express themselves, understand and relate to others and how they deal with everyday demands and pressures. | Bar-On Emotional intelligence model | (a) Quantitative (b) Structural equation model (predictive) (c) 958 | 1. Bar-On EQ-I | (1) Interpersonal problem solving (2) Self-actualization (3) Independent thinking (4) Stress management (5) Adaptability (6) Interpersonal relationship |

**Table A1.** *Cont.*

| ID | Citation | Participants Characteristics | Socioemotional Competence Definition/or Similar Concept | Theoretical Model | Approach, Design and Sample | Instruments | Instruments Dimensions |
|---|---|---|---|---|---|---|---|
| 5 | Chica et al. [91] | (a) Colombia (b) Not available (c) In service teachers | Emotional competence: the group of knowledge, capacities, abilities and attitudes necessary in order to understand, express and regulate the emotional phenomena in an appropriate way. | Not available | (a) Qualitative (b) Multiple case study (c) 156 | 1. Field journals of student practices and discussion groups 2. Open questionnaire | (1) Emotional conscience (2) Emotional regulation (3) Emotional autonomy (4) Social competencies (5) Competencies for life and wellbeing |
| 6 | Garner [92] | (a) USA (b) Primary-secondary (c) Pre/in service teachers | Not available | Not available | (a) Quantitative (b) Hierarchical regression analysis (associative) (c) 175 | 1. Subscale of Beran 2. Dyadic Trust Scale 3. Classroom Expressiveness Questionnaire | (1) Normative beliefs (2) Assertive beliefs (3) Avoidance beliefs (4) Dismissive beliefs (5) Prosocial beliefs (6) Empathy for victims (7) Mental representations of relationships (8) Confidence about managing bullying (9) Positive expressiveness (10) Negative expressiveness |
| 7 | Hen and Goroshit [93] | (a) Israel (b) Primary–Secondary (c) Inservice teachers | Not available | Not available | (a) Quantitative (b) Structural equation model (predictive) (c) 312 | 1. Self-Efficacy Scale 2. Inter-personal Reactivity Index | (1) Understanding (2) Perceiving (3) Facilitating (4) Regulating (5) Class context (6) School context (7) Fantasy (8) Empathic concern (9) Perspective taking (10) Gender (11) Academic degree (12) Years of work experience |

**Table A1.** *Cont.*

| ID | Citation | Participants Characteristics | Socioemotional Competence Definition/or Similar Concept | Theoretical Model | Approach, Design and Sample | Instruments | Instruments Dimensions |
|---|---|---|---|---|---|---|---|
| 8 | Hen and Sharabi-Nov [94] | (a) Israel (b) Primary (c) Inservice teachers | Emotional intelligence: refers to the ability to process emotional information as it pertains to the perception, assimilation, expression, regulation and management of emotion. | Emotional intelligence model. | (a) Quantitative (b) Quasi-experimental (c) 186 | 1. Interpersonal Reactivity Index (IRI) 2. Schutte Self Report Emotional Intelligence Test (SSREIT) 3. Reflection diaries | (1) Fantasy (2) Empathic concern (3) Perspective taking (4) Personal distress (5) Empathy (6) Expression of emotion (7) Regulation of emotion (8) Management of emotion (9) Emotional Intelligence |
| 9 | Karimzadeh et al. [95] | (a) Iran (b) Primary (c) Inservice teachers | Emotional intelligence: is an ability to identify and recognize the concepts and meanings of emotions, and their interrelationships to reason them out and to solve relevant problems. | Bar-On Emotional intelligence model | (a) Quantitative (b) Experimental (c) 68 | 1. Bar-On Social–emotional Questionnaire | (1) General mood (2) Adaptive ability (3) Interpersonal ability (4) Intrapersonal ability (5) Stress management |
| 10 | Knigge et al. [70] | (a) Germany (b) Secondary (c) Preservice teachers | Not available | Prosocial classroom model | (a) Quantitative (b) Experimental (c) 323 | 1. Self report 2. Interpersonal Reactivity Index | (1) Affective attitude behavioral (2) Affective attitude learning (3) Empathic concern (4) Perspective taking (5) Emotional exhaustion (6) Goal student–teacher relationship |
| 11 | Maiors et al. [96] | (a) Romania (b) Secondary (c) Inservice teachers | Social–emotional competencies include five core competencies: self-awareness, social awareness, self-management, relationship skills, and responsible decision making. | Social and emotional learning model | (a) Quantitative (b) Correlational (c) 81 | 1. Socioemotional competence questionnaire. | (1) Basic Needs Satisfaction (2) Rational Beliefs (3) Emotional Exhaustion (4) Depersonalization (5) Personal Accomplishments (6) Social emotional competencies |

**Table A1.** *Cont.*

| ID | Citation | Participants Characteristics | Socioemotional Competence Definition/or Similar Concept | Theoretical Model | Approach, Design and Sample | Instruments | Instruments Dimensions |
|---|---|---|---|---|---|---|---|
| 12 | Martzog et al. [55] | (a) Germany (b) Not available (c) Preservice teachers | Social–emotional competencies: multifaceted and include the teacher's ability to be self-aware, to be able to recognize their own emotions and how their emotions can influence the classroom situation. | Not available | (a) Quantitative (b) Quasi-experimental (c) 148 | 1. Interpersonal Reactivity Index IRI. | (1) Empathic concern (2) Perspective taking (3) Fantasy (4) Personal distress |
| 13 | Oberle et al. [56] | (a) Canada (b) Primary (c) Inservice teachers | Teacher SEC: a comprehensive set of interrelated skills and processes, including emotional processes, social and interpersonal skills, and cognitive processes. | Prosocial classroom model | (a) Quantitative (b) Associative, predictive model (c) 35 | 1. 6-item Students' Perceptions of Teacher Social–emotional Competence scale (TSEC) | (1) Teacher burnout (2) Classroom autonomy (3) School socioeconomic level (4) Age (5) Sex |
| 14 | Peñalva et al. [97] | (a) Spain (b) Not available (c) Preservice teachers | Emotional competence refers to the knowledge, capacities, abilities and attitudes that are considered necessary to understand, express and properly regulate emotional phenomena. | Not available | (a) Quantitative (b) Descriptive (c) 110 | 1. Emotional competence scale. | (1) Self-awareness (2) Self-regulation (3) Self-motivation (4) Empathy (5) Social skills |
| 15 | Pertegal-Felices et al. [98] | (a) Spain (b) Primary–Secondary (c) Pre/in service teachers | Emotional intelligence: is based on ability, aptitude, skill or efficiency that lead the person to a successful performance at work. | Not available | (a) Quantitative (b) Ex post facto comparative (c) 287 | 1. Traid Meta-Mood Scale-24 (TMMS-24) 2. Bar-On EQ-i:S. 3. NEO-FFI. | (1) Attention (2) Clarity (3) Repair (4) Intrapersonal intelligence (5) Interpersonal intelligence (6) Adaptability (7) Stress management (8) Humor (9) Emotional stability (10) Extroversion (11) Openness (12) Kindness (13) Responsibility |

**Table A2.** Extraction information matrix: theorical model of SEC.

| n° | n° Citation | Model | Core Concept | Description | Dimensions | Instruments | Use Population |
|---|---|---|---|---|---|---|---|
| 1 | 8926 | Emotional regulation process model [64] | Emotion regulation: is defined and distinguished from coping, mood regulation, defense, and affect regulation. Emotion is characterized in terms of response tendencies. | The emotion regulation process model facilitates the analysis of types of emotion regulation. This model has five sets of emotion regulatory processes: situation selection, situation modification, attention deployment, cognitive change, and response modulation. This is an elaboration of two-way distinction between antecedent-centered emotion regulation, which occurs before the emotion is generated, and response-centered emotion regulation, which occurs after the emotion is generated. | (a) Situation selection (b) Situation modification (c) Attentional deployment (d) Cognitive change (e) Response modulation | Emotion Regulation Questionnaire (ERQ) Dimensions: (a) Cognitive reappraisal, (b) Expressive suppression | Adults Children and Adolescents |
| 2 | 2935 | Prosocial classroom model [17] | Social and emotional competence: use the broadly accepted definition of social and emotional competence developed by CASEL (2008). This definition involves five major emotional, cognitive, and behavioral competencies: self-awareness, social awareness, responsible decision making, self-management, and relationship management. | The prosocial classroom mediational model establishes teacher social and emotional competence (SEC) and wellbeing as an organizational framework that can be examined in relation to student and classroom outcomes. Teachers' SEC and wellbeing influences the prosocial classroom atmosphere and student outcomes. This model recognizes teacher SEC as an important contributor to the development of supportive teacher–student relationships; teachers higher in SEC are likely to demonstrate more effective classroom management and they will implement a social and emotional curriculum more effectively because they are outstanding role models of desired social and emotional behavior | (a) Teacher's social–emotional competence and wellbeing (b) Teacher–student relationships (c) Effective classroom management (d) Social–emotional learning program implementation (e) Classroom climate | Different instruments for measuring SEC, example: 1. Interpersonal Reactivity Index 2. TSEC perception scale | Adults |
| 3 | 55 | Social and emotional learning model [26] | Social and emotional learning: involves the processes through which children and adults acquire and effectively apply the knowledge, attitudes, and skills necessary to understand and manage emotions, set and achieve positive goals, feel and show empathy for others, establish and maintain positive relationships, and make responsible decisions. | CASEL has identified five interrelated sets of cognitive, affective, and behavioral competencies: self-awareness, self-management, responsible decision making, relationship skills, social awareness (CASEL, 2013). The framework takes a systemic approach that emphasizes the importance of establishing equitable learning environments and coordinating practices across key settings to enhance all students' social, emotional, and academic learning. It is most beneficial to integrate SEL throughout the school's academic curricula and culture, across the broader contexts of schoolwide practices and policies, and through ongoing collaboration with families and community organizations. | (a) Self-awareness (b) Self-management (c) Responsible decision making (d) Relationship skills (e) Social awareness | Socioemotional competence questionnaire Dimensions: (a) Self-awareness, (b) Self-management, (c) Responsible decision making, (d) Relationship skills, (e) Social awareness | Adults Children and Adolescents |

**Table A2.** *Cont.*

| n° | n° Citation | Model | Core Concept | Description | Dimensions | Instruments | Use Population |
|---|---|---|---|---|---|---|---|
| 4 | 2105 | Bar-On Emotional Intelligence model [65] | Emotional intelligence: is an array of noncognitive capabilities, competencies, and skills that influence one's ability to succeed in copying with environmental demands and pressures. | The Bar-On model provides the theoretical basis for the EQ-i, which was originally developed to assess various aspects of this construct as well as to examine its conceptualization. According to this model, emotional–social intelligence is a cross-section of interrelated emotional and social competencies, skills and facilitators that determine how effectively we understand and express ourselves, understand others and relate with them, and cope with daily demands. | (a) Intrapersonal skills (b) Interpersonal skills (c) Adaptability (d) Stress management (e) General mood | Bar-On EQ-I Dimensions: (a) Intrapersonal, (b) Interpersonal, (c) Stress management, (d) Adaptability, (e) General mood | Adults Children and Adolescents |
| 5 | 12,606 | Emotional intelligence model [25] | Emotional intelligence: is a set of abilities that account for how people's emotional perception and understanding vary in their accuracy. More formally, we define emotional intelligence as the ability to perceive and express emotion, assimilate emotion in thought, understand and reason with emotion, and regulate emotion in the self and others. | The model considers that emotional intelligence is conceptualized through four basic skills: the ability to accurately perceive and express emotions, the ability to access and/or generate feelings that facilitate thought; the ability to understand emotions and emotional awareness and the ability to regulate emotions promoting emotional and intellectual development. | (a) Perception and expression of emotion (b) Assimilating emotion in thought (c) Understanding and analyzing emotion (d) Reflective regulation of emotion | Self-report measure: Trait Meta-Mood Scale Dimensions: (1) Attention (2) Clarity (3) Repair Performance measurement: MSCEIT Dimensions: (1) Perceiving and expressing emotions (2) Using emotions, (3) Understanding emotions, (4) Regulating emotions | Adults Children and Adolescents |

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
