# Peer review of "Teachers’ Social–Emotional Competence: History, Concept, Models, Instruments, and Recommendations for Educational Quality"

_sustainability, doi:10.3390/su132112142_

Round 1
Reviewer 1 Report
Dear Authors,
I really appreciate your work.
However, I have some concerns about it. For example, page 3, there are the following sentences:
"3.2. Theoretical SEC Models and Instruments
3.2.1. Theoretical SEC models
Five theoretical models that met the inclusion criteria have been selected to be analyzed in the present study: Gross' model of the emotional regulation process in 1998 [58]; Mayer and Salovey’s emotional intelligence model in 1997 [25]; Bar On's emotional intelligence model in 1997 [59]; Jennings and Greenberg's 2009 prosocial classroom model [17]; and the CASEL social-emotional learning model in 2013 [26]. The models are characterized below
Emotion Regulation Process Model
Gross' model is based on emotional regulation, understood as the processes by which people influence the emotions they have, and how they experience and express them. These emotion regulation processes can be automatic or controlled, conscious or unconscious, and their effects can be shown at one or more points in the emotion generation process [60].
This emotion regulation process model facilitates and allows the analysis of types of emotion regulation by establishing five sets of emotional regulatory processes as shown in figure 5: situation selection, situation modification, attention deployment, cognitive change, and response modulation. This is an elaboration of the two-way distinction between antecedent-centered emotion regulation, a pre-emotion process, and response-centered emotion regulation, a process that occurs after the emotion is generated"
Thus, the concept of SEC is comparable to the Emotion Regulation Process Model? is the answer is affirmative, why didn’t you do a search using these (and others) terms?
There is literature about the emotion regulation process in teachers.
See for example:
https://scholar.google.com/scholar?hl=en&as_sdt=0%2C5&q=emotion+regulation+process+in+teachers&btnG=
(this is a search in google scholar with the terms "emotion regulation process in teachers").
I hope to have the opportunity to revise your work.
Author Response
Dear reviewer,
We are writing in relation to the manuscript entitled "Teachers´ social-emotional competence: history, concept, models, instruments, their relevance in educational quality, and recommendations for their development". Authors: Gissela Lozano-Peña, Fabiola Sáez-Delgado, Yaranay López-Angulo and Javier Mella-Norambuena.
We would like to thank you for your comments, which we believe have greatly improved the quality of our article. We have modified the text accordingly and responded to your comment.
Response: The answer is negative; SEC is not the same as emotional regulation. The purpose of this review was to identify the models and instruments used in empirical research to work on SEC. For this reason, the search was carried out with this concept (SEC). Now, some researchers in these studies, only focus on the aspect of emotion regulation using theoretical models; even when their study was presented under the construct of SEC. Undoubtedly, SEC and emotion regulation are different constructs. Emotion regulation is part of SECs according to several theoretical models; therefore, it is more specific.
The difference between SEC and emotional regulation is stated in the literature. The authors point out that social and emotional competencies are considered inseparable constructs because the social development of teachers has an inherent emotional component since the teacher must learn to express and regulate emotions in order to participate successfully in social interactions. The socioemotional competence of teachers is therefore composed of a set of skills necessary to face social and emotional expectations of the educational context (King & La Paro, 2018). Therefore, research is clear in recognizing that socially competent behavior is characterized by two aspects: on the one hand, social competence, which is manifested effectiveness in social interactions. On the other hand, emotional competence, which is manifested in culturally accepted emotional expression, reflective understanding of one's own and others' emotions, and effective emotion regulation (Rose-Krasnor & Denham 2009; Silkenbeumer, Schiller, Holodynski, & Kärtner, 2016).
Therefore, models of emotional regulation were not included in the search. However, the reviewer's observation was considered important, and the discussion and conclusion were improved.
Although this was pointed out in the discussion noting that some models are limited to only some aspects of socioemotional competence (see L510-520), as follows:
The results of this study also evidenced the existence of 5 theoretical models that empirical research uses as a theoretical basis in studies that address SEC with a sample of school-level teachers. Specifically, two of these, corresponding to the Bar On’s and Mayer and Salovey’s models, are directly supported by the theory of emotional intelligence, since this is the central concept [25], therefore, they were classified as models of emotional intelligence. In relation to Jennings and Greenberg’s prosocial classroom, and that of CA-SEL social-emotional learning, are more inclusive, since they consider learning and/or social and emotional skills to be central concepts and include social and emotional skills which are called integrative skills, therefore, these two were classified as social and emotional development models. Finally, Gross’ emotional regulation Model was focused on the emotional dimension of SEC.
Therefore, in the conclusion it was added that future studies propose complex models that integrate all teachers SEC dimensions (Social, emotional, competence), as follows:
In conclusion, this study contributes to academia and education. In the academic aspect, this study presents to researchers the historical path of SEC concept and its configuration from different theoretical approaches showing its evolution. It opens the field for the proposal of new theoretical models focused on how to develop the three-dimensionality of the SEC concept, framing it specifically to the research field of teachers.
Reference:
King & La Paro. (2018). Teachers’ Emotion Minimizing Language and Toddlers’ Social Emotional Competence. Early Education and Development, 29(8), 989-1003.https://doi.org/10.1080/10409289.2018.1510214
Silkenbeumer, Schiller, Holodynski, & Kärtner. (2016). The role of co-regulation for the development of social-emotional competence in early childhood. Journal of Self-Regulation and Regulation, 2, 11–26. https://doi.org/10.11588/josar.2016.2.34351
"Please see the attachment."
Yours Sincerely,
Authors

Reviewer 2 Report
Dear Authors,
Thank you very much for submitting your text to "Sustainability". I have read it with great interest.
Its advantages are the subject matter and the pro-social tone, but not only. I think that your paper is based on a well-thought-out structure. A consistent narrative and logical argument characterise it. The text is clear, reliable and of high quality. It was an absolute pleasure to read it, and I thank you very much for that.
However, I noticed a few minor flaws, the removal of which could have improved the value of your study. I list them below.
- I suggest shortening the title to make it easier to read;
- in the "Introduction", you can identify the academic gaps observed in your research (lines 86-87) more clearly. At the moment, I think that the argumentation in the introduction does not allow for a clear identification of these gaps;
- in section 2.1.2, it is unclear why the exclusion criterion was articles with theoretical research (i.e. why did you focus only on the empirical aspect)? It is confusing, mainly since you then focus on theoretical models. It would be worthwhile to justify and explain this issue;
- also, in section 2.1.2, under the caption Figure 1, you write about "records excluded, without teachers' SEC's concept in the title or abstract". What about keywords, which you also (line 120) indicated as a criterion for text selection?
- Was the criterion for selecting texts their form (i.e. only articles or chapters in books)?
- How did you analyse the final sample of 15 texts? Did you use a quantitative-qualitative analysis and a specially developed coding key? Was the whole team or members of the team involved? Did you cross-check the analyses? It would be helpful to explain the process of not only aggregation but also analysis of the data;
- it is not clear why in Figure 3 (and in this part of the analysis) you refer to texts outside the period you have chosen (i.e. 2010-2021);
- among the limitations of the study, I would also point out the small research sample and the period of analysis (10 years, although there were some texts dedicated to this research area earlier);
- in "Conclusions", I would emphasise the study's contribution to the academy more strongly, i.e. how it responds to academic gaps. In addition, I would highlight its contribution to education in the broader sense, as well as to the improvement of social life;
- among the indications for future research, I would refer to the pandemic - this thread is missing here, and I can assume that it has a considerable impact on teachers' SECs;
- The Spanish language appears in some places (in tables, figures).
These are, of course, only some recommendations and I hope you will consider them. Good luck!
Sincerely.
Author Response
We appreciate the reviewer's suggestions. Thank you for your comments. All improvements were incorporated into the manuscript. Please review the detailed responses in the cover letter.

Round 2
Reviewer 1 Report
Dear Authors, I appreciate your effort to improve the manuscript.